# Extensive spatial impacts of oyster reefs on an intertidal mudflat community via predator facilitation

Carl J. Reddin [1,2✉], Priscilla Decottignies[1], Lise Bacouillard[3], Laurent Barillé [1], Stanislas F. Dubois[3], Caroline Echappé [1], Pierre Gernez [1], Bruno Jesus[1], Vona Méléder [1], Paulina S. Nätscher [4], Vincent Turpin[1], Daniela Zeppilli [5], Nadescha Zwerschke [6], Anik Brind'Amour[7] & Bruno Cognie [1]

Habitat engineers make strong and far-reaching imprints on ecosystem processes. In intertidal mudflats, the dominant primary producer, microphytobenthos (MPB), often forms high biomass patches around oyster reefs. We evaluate multiple hypotheses linking MPB with oyster reefs, including oyster biodeposition, meiofaunal grazing, and abiotic factors, aiming to help predict effects of reef removal or proliferation. We quantify spatial patterns of an Atlantic mudflat community and its environment around two large *Crassostrea* reefs before experimentally sacrificing one reef via burning. MPB biomass was enriched surrounding living oyster reefs although infaunal biomass and individual sizes were low. Structural equation modelling best supported the hypothesis that crab predation intensity, which decayed with distance from the reefs, locally freed MPB from grazing. Our results suggest that *Crassostrea* reef expansion may enrich local MPB patches and redirect trophic energy flows away from mudflat infauna, with potential implications for the sustainability of local fisheries and bird conservation.

[1] Université de Nantes, Département des Sciences de la Vie, EA 2160 Mer - Molécules - Santé 2, Rue de la Houssinière, 44322 Nantes, France. [2] Museum für Naturkunde - Leibniz Institute for Research on Evolution and Biodiversity, Invalidenstr. 43, 10115 Berlin, Germany. [3] French Research Institute for the Exploration of the Sea (IFREMER), DYNECO-LEBCO, Centre de Bretagne, ZI de la pointe du Diable, CS 10070, 29280 Plouzané, France. [4] GeoZentrum Nordbayern, Department of Geography and Geosciences, Universität Erlangen-Nürnberg, Erlangen, Germany. [5] French Research Institute for the Exploration of the Sea (IFREMER), REM-EEP-LEP, Centre de Bretagne, ZI de la pointe du diable, CS10070, 29280 Plouzané, France. [6] Joint Nature Conservation Committee, Inverdee House, Baxter Street, Aberdeen AB11 9QA, UK. [7] French Research Institute for the Exploration of the Sea (IFREMER), PDG-RBE-EMH, Rue de l'Ile d'Yeu, BP 21105, 44311 Nantes, France. ✉email: Carl.reddin@mfn.berlin

The establishment of novel ecosystem engineers often produces a cascade of positive and negative abiotic and biotic effects across the local community[1,2], which are often indirect and detectable tens of metres away from their structure[3–5]. Some cultured species successfully establish wild populations, which may have a marked impact on the recipient community, including changing the dynamics of nutrient and energy flows[6] or altering habitat physical structure[7]. One such marine species is the Pacific oyster, *Crassostrea gigas* (*Magallana gigas*) (Thunberg, 1793)[8–10], being introduced to 19% of the world's marine ecoregions[11], where it can form extensive, hard and complex reefs[12] that are of interest for nature-based coastal defense[13]. In North Atlantic tidal mudflats, *C. gigas* aquaculture and wild reefs alike facilitate persistent, high microphytobenthos (MPB) patches (diameter often >100 m) but the processes underlying this development and their impacts are not fully understood[14–18]. Meanwhile, warming waters and increased phytoplankton is expected to increase the extent of European *C. gigas* reefs[19].

Mudflat ecosystems are highly dynamic in spatial, temporal and ecological terms and provide reliable ecosystem functioning, services and goods[10,20], including shellfish aquaculture, fish and bird biomass[13,21,22]. The main primary producer, MPB, is usually comprised by diatoms, euglenids, and cyanobacteria living in the top few millimetres of mudflat sediment and can reach high levels of primary production[23–26]. MPB biomass can reach 200–300 mg chl a m$^{-2}$ but is highly variable in space, following elevation, tidal exposure and granulometry[15,27,28]. Hypotheses for high biomass patch formation include local nutrient enrichment from bivalve biodeposits (pseudofeces and feces) or the alteration of local current dynamics[14,29,30], while other authors emphasise the role of patchy distributions of benthic micro-grazers, e.g.,[31]. MPB is typically the main food source for mudflat benthic macrofauna[5], which are often mobile or suspension feeding to efficiently assimilate MPB's patchy distribution. The importance of MPB has been demonstrated in both benthic and pelagic, temperate and subtropical, intertidal and subtidal food webs, which in turn support higher trophic levels[23,32–34]. This leads to a positive bottom-up relationship between MPB biomass and its macrofaunal consumers (i.e., obligate and facultative surface deposit and suspension feeders)[35]. Cultured shellfish also consume MPB,

but oysters may also promote MPB growth through a feedback mechanism based on excreted mucus[36]. Therefore, commercial outputs of both primary consumers, such as shellfish farms, and fisheries of secondary consumers, such as the flatfish *Solea solea*, could be sensitive to changes in MPB primary production[22,37]. Given its importance, the understanding of how spatial variation in MPB biomass is related to consumer biomass remains relatively poor.

Marine ecosystem engineers may also encourage the top-down regulation of communities, both by their own feeding activity and by supporting additional consumers[2,38]. For example, habitat provision by coral reefs may encourage intense grazing for tens of metres around the reefs, leaving bare 'halos' denuded of macroalgae[39]. Oyster reefs may function similarly with implications for food web structure, including indirect facilitation or suppression of meio- and macrofauna primary consumers. These in turn may facilitate or suppress MPB depending on the trophic level facilitated, which may alter competitive dynamics with native suspension feeders for space and food[8,40]. Anthropogenic impacts on reefs may be sudden, as oyster farmers look to harvest wild stocks, especially to replenish cultured stocks following disease-associated spat failures[17].

Despite their importance for ecosystem goods and services, mudflats are challenging to sample, although high-resolution images of the system are available by remote sensing[16,17]. Ecological processes may also vary in relative importance with distance from the reef, can be intertwined and dynamic in nature, and are best investigated by combining the strengths of correlative and experimental study. We sampled mudflats of Bourgneuf Bay, northwest France, 360° in the near (10 s of m) and far (100 s of m) surroundings of two oyster reefs, each >750 m$^2$, over two seasons for sediment, MPB, meiofauna and macrofauna on a regular spatial grid (196 sample stations, Fig. 1). To identify which process depends on living oysters, we sampled before and after a large before-after-control-impact (BACI) experiment, sacrificing the oysters of one reef while leaving the structure intact via burning while maintaining the other reef as a control. Supported by a 30 year time series of remotely sensed multispectral images for MPB estimation[16], we used structural equation modelling (SEM) and geostatistics to compare evidence for four

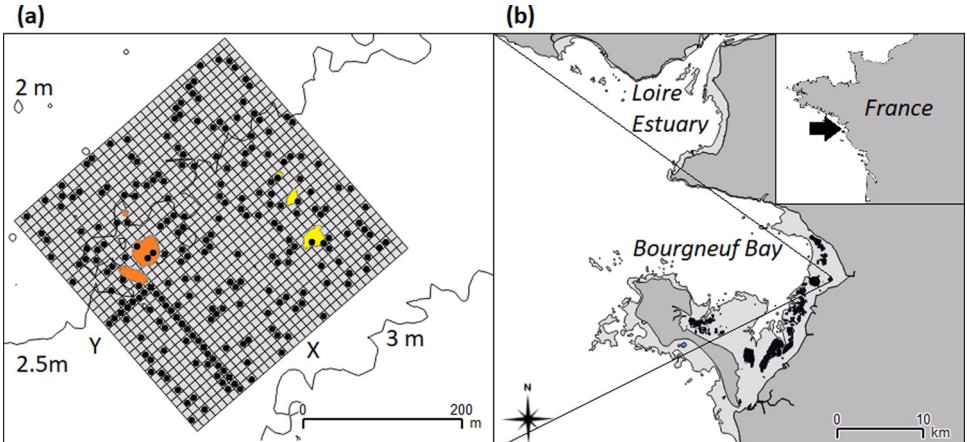

**Fig. 1 Location of the study grid within Bourgneuf Bay covering two groups of oyster reefs (47°01′32″ N, 2°00′26″ W). a** Shows the control (orange) and treatment (yellow) reefs and the superimposed study grid (edge length 350 m, individual cell edge 10 m) with selected cells for core sampling (black dots), bathymetry (contours), and '*X*' and '*Y*' axes of the grid labelled. Grid *y*-axis coordinates approximate bathymetry, Pearsons r = −0.87). We used the inverse 'L' shape of samples near the southern corner of the grid for in situ quantification of MPB pigment composition and biomass to complete the remotely sensed MPB biomass estimates (NDVI; see Supplementary Results: Groundtruthing MPB for details). **b** Shows grid location in Bourgneuf Bay, showing intertidal areas (light grey), mainland (dark grey) and oyster farms (black). Inset panel shows location of Bourgneuf Bay (arrow) in the North Atlantic coastline of north-west France. Further illustrations in Echappé et al.[16].

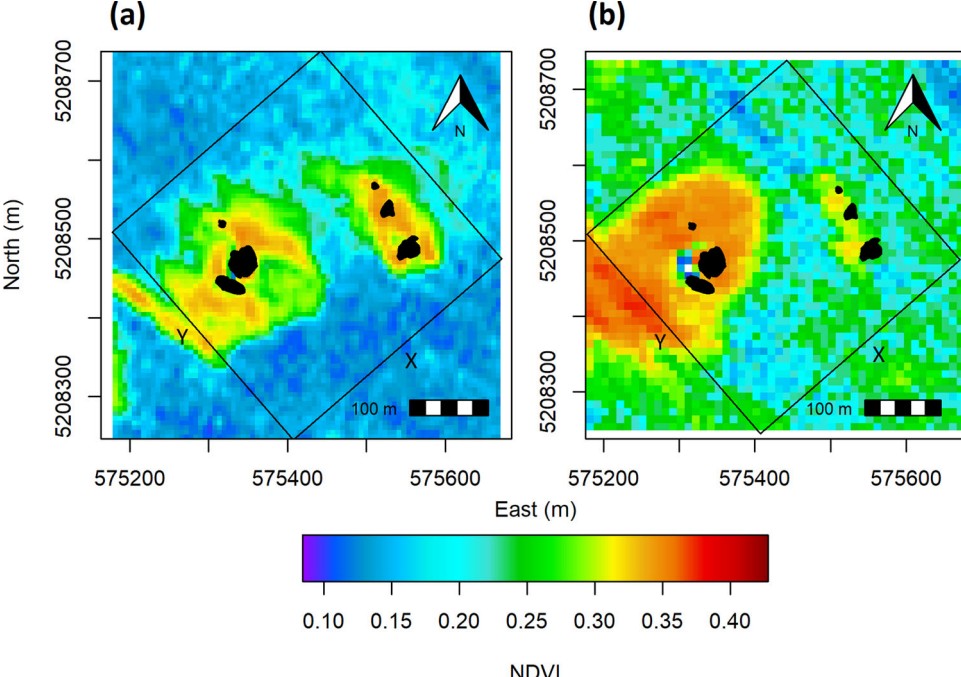

**Fig. 2 MPB biomass over the study extent via the remotely sensed proxy of NDVI. a** Shows late summer 2013, before experimental treatment (SPOT6 image on 20 August, 10:46 h, 1.75 h after low tide, at 6 m resolution), and **b** shows early autumn 2014, after experimental treatment (SPOT5 image on 9 October, 10:22 h, approximately at low tide, at 10 m resolution). NDVI minima and maxima approximately translate to <50 and >250 mg Chl $a$ m$^{-2}$ (Supplementary Fig. S1). Further before-after images are available in Supplementary Figs. S4–7. Black square shows the extent of field-sampled grid, with $x$- and $y$-axes labelled. Filled black contours show the extent of oyster reefs. Image NDVI 2013 median = 0.12; 2014 median = 0.15. Grid coordinates in UTM system, projection WGS84 UTM30.

hypothetical, non-mutually-exclusive processes to regulate MPB and structure the community. The hypothesised processes were:

(1) 'Oyster biodeposition' delivers nutrients, faeces, pseudo-faeces and mucus, causing local enrichment of MPB biomass[16].
(2) 'Abiotic': the physical effect of the reef alters tidal currents to locally concentrate MPB biomass[29].
(3) 'Predation', primarily by facilitated crabs[6], concentrates around the reef and provides top-down control of macrofaunal grazer biomass and body sizes, locally releasing MPB from grazing pressure.
(4) 'Meiofauna' grazing pressure on MPB is lower near to the reefs, permitting higher MPB biomass[38].

### Results
**BACI patterns of microphytobenthos, granulometry and organic matter.** During early autumn before treatment, NDVI was very strongly, negatively correlated with distance from either oyster reef ($r = -0.80$, df $= 20.1$, $n = 196$, $P < 0.001$; Fig. 2a). NDVI patterns were validated as representing MPB biomass both before and after the treatment (Supplementary Results: Ground truthing MPB, Supplementary Fig. S1–S5, Supplementary Table S1). NDVI was highest around the large western control reef ($r = -0.51$, df $= 16.7$, $n = 196$, $P = 0.028$), while the smaller treatment reef produced a smaller NDVI patch ($r = -0.18$, df $= 20.9$, $n = 196$, $P = 0.401$). In winter, NDVI was unrelated to distance from the reefs (Supplementary Results: Additional MPB Images, Supplementary Figs. S4 and S6).

During early autumn post-treatment, we observed a relative decrease in NDVI at the treatment reef and a relative increase at the control reef (Fig. 2b). A temporal series of NDVI images suggested that this decrease was gradual, being most visible from

two months after treatment (Supplementary Results: MPB Time Series, Supplementary Fig. S7; also[16]). MPB biomass increased across and outside the study grid from before to after the treatment, 2013 to 2014, although the 2014 NDVI mean (0.163) remained within the bounds of long-term variation for the season of 0.16 ± 0.02 (mean ± SD; Fig. 8 in[16]). Both shallow sediment % organic matter (OM) and median grain size (MGS) increased from 2013 to 2014 (Table 1; Supplementary Figs. S8), the former also concentrating to the seaward side of the reefs. Major detected pigments (Chl $c$, Fuco, DD and DT) and their respective ratio to Chl $a$ indicated that diatoms dominated MPB both before and after the treatment. Other pigments (Chl $b$, Zeaxanthin, Lutein and Pheophytin $b$) were not detected, or only in trace concentrations. However, pigment ratios to Chl $a$, including carotenoid by-products, decreased from before to after the treatment, indicating that diatoms (usually rich in carotenoids) decreased in their proportion of total MPB biomass, though still dominated. Meanwhile, pheophorbid $a$/Chl $a$ increased, indicating a relative increase in rates of grazing e.g., by *P. ulvae* (Table 1; Supplementary Figs. S2 and S3).

**Oyster biodeposition controls MPB.** Potential support for the *biodeposition* hypothesis (1) was observed as direct positive relationships from reef proximity to NDVI independent of the indirect pathways permitted in the structural equation models (i.e., via macrofaunal or meiofaunal grazing; Fig. 3). However, positive relationships were observed both pre-treatment and after the treatment oysters were sacrificed (also Fig. 2a, b), with MPB not completely collapsing near to the treatment reef. Thereby, this direct pathway and hypothesis retained a stronger correlation in 2014, though with a high variance, than correctly assuming mortality of the treatment reef (Fig. 3c vs b). This suggests that

**Table 1 Pre- (2013) and post-treatment (2014) MPB biomass, pigment composition and sediment conditions from early autumn.**

| | 2013 | | | 2014 | | | Wilcoxon's test | |
|---|---|---|---|---|---|---|---|---|
| | Median | MAD | n | Median | MAD | n | V | P |
| Biomass (mg Chl $a$ m$^{-2}$) | 80.46 | 43.62 | 66 | 126.81 | 54.73 | 66 | 1301 | <0.0001 |
| Fuco/Chl $a$ (%) | 31.01 | 5.22 | 66 | 22.03 | 3.62 | 66 | 1027 | <0.0001 |
| DD + DT/Chl $a$ | 4.99 | 1.86 | 66 | 3.52 | 0.73 | 66 | 877 | <0.0001 |
| Chl $c$/Chl $a$ | 5.33 | 1.17 | 66 | 4.42 | 0.99 | 66 | 805 | <0.0001 |
| Carotenoid by-products/Chl $a$ | 9.63 | 2.83 | 66 | 6.71 | 1.64 | 62 | 955 | <0.0001 |
| Pheophorbid $a$/Chl $a$ | 0.57 | 0.41 | 63 | 1.78 | 1.00 | 66 | 590 | <0.0001 |
| Shallow OM (%) | 8.0 | 2.9 | 190 | 11.7 | 3.2 | 132 | 736.5 | <0.0001 |
| Shallow median grain size (µm) | 35.6 | 9.2 | 182 | 40.1 | 15.4 | 149 | 1730 | 0.0001 |
| Deep OM (%) | 9.8 | 3.0 | 172 | 8.9 | 5.0 | 133 | 1330.5 | 0.34 |
| Deep median grain size (µm) | 52.3 | 23.1 | 143 | 44.1 | 17.3 | 186 | 3172 | 0.07 |
| $M.\ balthica$ biomass (AFDW g m$^{-2}$) | 0.77 | 0.76 | 190 | 0.5 | 0.42 | 187 | 12697 | <0.0001 |
| $S.\ plana$ biomass (AFDW g m$^{-2}$) | 0.07 | 0.10 | 190 | 0 | 0 | 187 | 5348 | 0.31 |
| $P.\ ulvae$ biomass (AFDW g m$^{-2}$) | 0.26 | 0.25 | 190 | 0.21 | 0.17 | 52 | 758 | 0.15 |

Medians compared using paired Wilcoxon's test. For spatial variability see Supplementary Figs. S1–3, S8–15.
MAD median absolute deviation.

mechanisms additional to oyster biodeposition were correlated with these paths (see below).

**Abiotic control of MPB**. With sediment and bathymetry variables being uncorrelated with NDVI at the scale of our study grid, the *abiotic* hypothesis (2) was unsupported during early autumn, both pre- and post-treatment (Fig. 3, Supplementary Figs. S9 and S10). Instead, abiotic variables were more important during winter (Supplementary Results: Additional MPB Images, Fig. S4).

**Predation release of MPB**. The pathway from distance from living reef to grazer biomass, here hypothesised to relate to top down control by predation, remained positive from before treatment to after treatment (Fig. 3a–c). Around the control reef, post-treatment MPB remained high (Fig. 2b), suggesting that predation control of grazers remained intact, but not around the treatment reef (e.g., sign reversal of NDVI~grazer path coefficient in Fig. 2b). This fits a scenario where treatment reef predators perished in the burning of the treatment reef and generally were not replaced, which allowed higher grazing on the treatment reef than before the treatment.

Reef proximity had a large impact on mudflat biomass but not on diversity (Fig. 4, Supplementary Fig. S11, Supplementary Table S2). Three macrofaunal species, the prey under this hypothesis, comprised 88.0% of the total pre-treatment abundance (Supplementary Table S3): the epifaunal grazer *P. ulvae* and the infaunal facultative suspension feeders *M. balthica* and *S. plana*. Pre-treatment biomass estimates for the 0.1225 km$^2$ study grid for *M. balthica* and *S. plana* were in the same order of magnitude as the combined *C. gigas* reefs (Supplementary Table S4). Biomasses of *M. balthica*, *P. ulvae*, *S. plana*, and the predatory polychaete *Nephtys hombergii*, also likely to be preyed upon, all showed similar distance decay around the oyster reefs (Fig. 4, Supplementary Figs. S12–15, Table 2, Supplementary Table S2, Supplementary Results: Macrofauna Details) via decreases in both abundance and mean individual size (Supplementary Fig. S16). A slight decrease in pre-treatment diversity of mudflat fauna via Simpson's D with distance from the reefs was of marginal significance (Fig. 4, Table 2). Spatial correlations between MPB and grazer biomasses were strong and negative (Table 2), supporting grazing pressure on MPB. *S. plana* and *N. hombergii* biomasses were weak-moderately correlated with MPB and *S. plana* was more strongly correlated with distance from the oyster reefs than with MPB (Table 2). No infaunal species'

biomass was significantly correlated with MGS over our study extent. Overall biomass of *M. balthica* decreased significantly post-treatment, while overall biomass of the more mobile *P. ulvae* showed no significant change (Table 1). Post-treatment biomass of the deeper-digging *S. plana* effectively fell to zero, although a strong link to bathymetry suggested that the seaward side of the plot changed more post treatment.

Fyke net and reef quadrat sampling over high tide identified the crab *C. maenas* as the most abundant and likely predator of infaunal bivalves (Supplementary Table S5). Three individuals of the flatfish *Solea solea* were also caught but their intertidal foraging is unlikely confined to the reef surroundings[22]. *C. maenas* mean carapace width, which integrated both abundance and body size, decreased with distance from the reefs (R = −0.53, $P = 0.08$, df. = 9.99, $n = 20$; Fig. 5). Linear extrapolation from the significant regression in Fig. 5 suggests that at 65 m from the oyster reefs, approximately where body sizes of *P. ulvae* and *M. balthica* both increased to a plateau (Supplementary Fig. S16), crab mean carapace width would be just 15 mm (95% confidence intervals = 4.1–25.8 mm). Mean body size (g) of pre-treatment *P. ulvae* became larger with distance from both reefs (Rho = 0.272, $x$ axis detrended), which was a stronger correlation than with MPB (Rho = −0.214, $x$ axis detrended; Supplementary Fig. S16). Meanwhile, size variance among *P. ulvae* individuals decreased with distance from the reefs (Rho = −0.549) and with decreasing MPB (Rho = 0.586; Supplementary Fig. S16). *M. balthica* did not share this pattern in size variance (with distance from reefs, Rho = 0.15) but mean body sizes became smaller (Rho = 0.234) in close proximity to both reefs (distances < 60 m, Rho = 0.295; Supplementary Fig. S16). We observed conspicuous large crabs in the reef rockpools, mostly *C. maenas* but also *Hemigrapsus sp.*, including *H. takanoi* (maximum carapace widths, *C. maenas* 110 mm and *H. takanoi* 50 mm). The only other abundant predator caught in the nets was the shrimp *C. crangon*, the abundance of which was unrelated to distance from reefs (Rho = −0.12, $n = 20$, $P = 0.54$; Supplementary Results: Reef Epifauna Details).

**Meiofauna controls MPB**. Although the SEM path from reef distance to nematodes was significant ($P = 0.02$), neither abundances of copepods or nematodes, the two most dominant meiofauna, were correlated with MPB (Fig. 3, Supplementary Fig. S17, Table 2). Additionally, the direct path from reef distance to NDVI was highly significant, alluding to mechanisms independent from

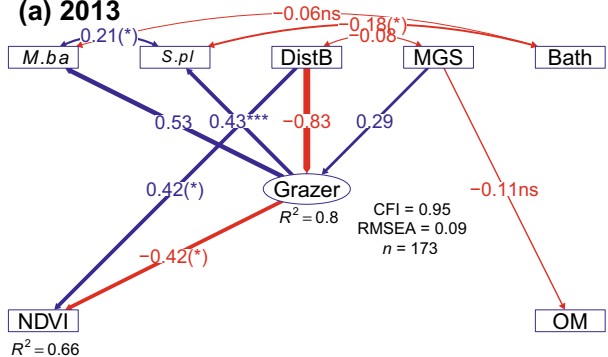

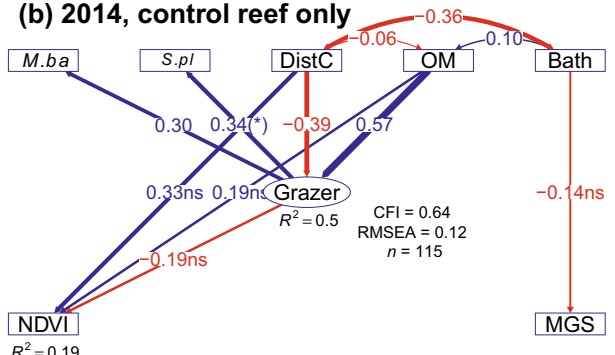

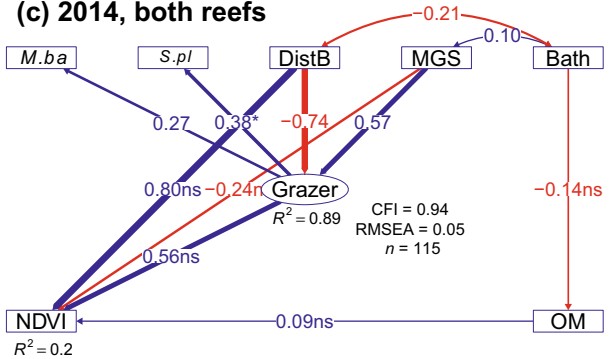

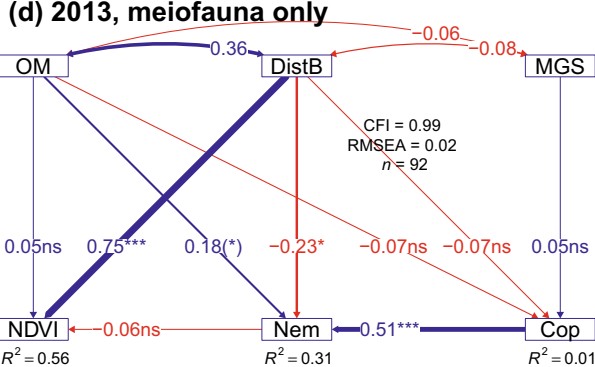

**Fig. 3 Path diagrams showing the change in spatial patterns in MPB and other variables, from before to after the sacrifice of the treatment reef via burning. a** and **d** show before (2013), models based on $n = 173$ and 92, respectively; **b** and **c** show after (2014), models based on $n = 115$ for each (see methods for details). **b** assumes the burning nullified the effect of the treatment reef (i.e., under hypothesis 1 or 3), while **c** assumes no post-burning change in the effect of the treatment reef (i.e., under hypothesis 2). Inset legend shows model fit statistics. Paths between variable nodes and their edge width show the standardised correlation coefficients (β), the effect of a variable in isolation (other variables held constant). Paths weaker than $\beta = \pm 0.05$ are not displayed. Sample sizes and standard errors of endogenous variables were corrected for spatial autocorrelation and used to calculate significance levels, denoted by asterisks: *$p < 0.05$, **$p < 0.01$, ***$p < 0.001$. One-headed arrows depict hypothesised causal relationships. Two-headed arrows depict residual covariance. The circular node shows the latent variable 'Grazer', representing top-down effects of macrofaunal grazers. Square nodes denote directly measured variables, including organic matter ('OM'), NDVI, sediment median grain size ('MGS'), bathymetry ('Bath'; roughly a y-trend), biomasses of *M. balthica* ('M.ba') and of *S. plana* ('S.pl'), abundances of Nematoda ('Nem') and Copepoda ('Cop'), and the square root of distance from the control reef (DistC) or from both reefs (DistB; highest values are nearest to the reefs). For clarity, residual variances are not displayed.

meiofauna underlying this path. However, an indirect positive pathway from oyster reefs to nematode abundance via OM is suggested, even if the direct, negative relationship suggests other mechanisms inhibiting nematodes.

## Discussion

We provide a high spatial resolution evaluation of the impacts of a non-native habitat-engineer on a mudflat community. Analysis

to compare before and after an experimental intervention chiefly implicated the role of the *C. gigas* oyster reef to locally intensify crab predation. Crab predation is likely responsible for the wide spatial imprint on the macrofaunal biomasses and size structure, providing local grazing relief to MPB, which was observable as a halo of NDVI, maintained around the control reef throughout. MPB did not recover its high density around the sacrificed treatment reef, a statistically significant deviation from the preceding 25 years[16]. This deviation outside of variation caused by other environmental changes typical for the locality implicates an impact from the burning of the oyster reef. However, it was not clear if this deviation was due to a persistent absence of crabs, or the removal of a facilitation mechanism of local MPB densities by living oysters, such as being fertilised by oyster biodeposition[16]. Although some artefacts of this large experimental manipulation may have remained in the ecosystem sampled in 2014, our multifaceted results were mostly inconsistent with large experimental impacts besides the intended killing of oysters (Supplementary Methods: Details of the reef burning). Abiotic variables, such as a hydrological effect of the reef[29], may take a foremost role in winter in setting MPB patterns, and a deferential role in summer/early autumn, following elevated biological activity levels. Balance between these processes thus likely changes depending on season, local conditions and bathymetry[27].

MPB was negatively correlated with mudflat macrofauna biomass patterns of most dominant species, irrespective of the species' expected trophic dependence on MPB. Species with distributions correlated to MPB included *P. ulvae*, an obligate consumer of MPB, and *N. hombergii*, a predator of small crustaceans and polychaetes, suggesting that something other than MPB was the foremost control on macro-infaunal biomass patterns. However, negative correlations between MPB and biomasses of *P. ulvae* and *M. balthica*, both being major consumers of MPB[34], likely indicated that these grazers exerted control on MPB distribution[5,41]. The potential influence of grazing is demonstrated by the addition of *P. ulvae* to microcosms leading to the decrease in MPB abundances to 20–30% that of ungrazed controls[42]. Benthic feeding can also be the primary mode for *M. balthica* (diet >75% benthic diatoms) and *S. plana* (>60%)[43]. Grazing pressure on MPB may be highest from March

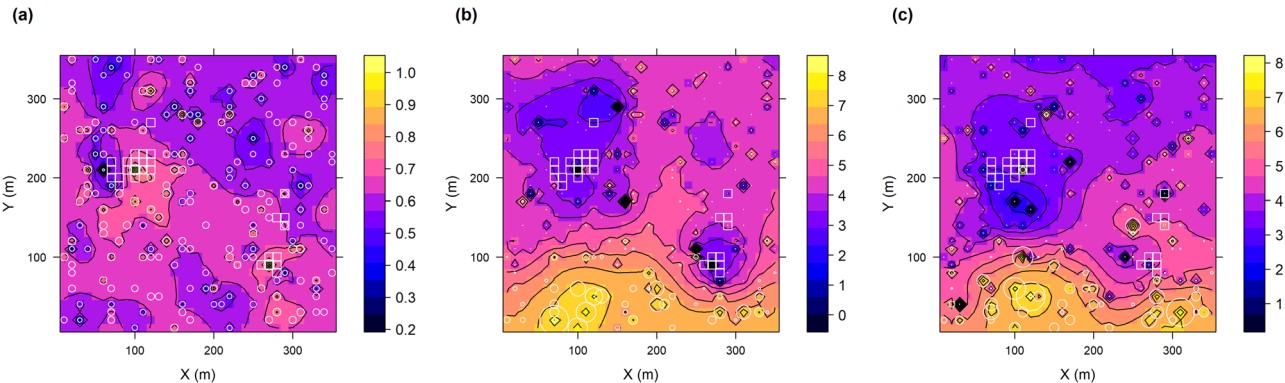

**Fig. 4 Kriging heat and contour maps of 2013 pre-treatment macrofaunal diversity and biomass around the oyster reefs.** (**a**) Shows diversity (Simpson's D) and (**b**) the summed biomass in AFDW grams per cell of the dominant bivalves *M. balthica*, and *S. plana*. For comparison, **c** shows the summed biomass of *M. balthica*, and *S. plana* after the treatment. White 10 * 10 m squares are those >50% occupied by oyster reef. All use simple Kriging prediction, which incorporates a bathymetric trend ('Y' axis) in these data, for interpolation of the sampled points (white circles, radius relative to sample value). Biomass data and scale bars are natural log transformed. See Supplementary Fig. S11 for prediction variance maps and individual species biomass maps. For 2014, *M. balthica*, and *S. plana* biomass samples were complete but sampling issues for other species precluded a comparable diversity plot.

to June in Bourgneuf Bay[28]. Thus, the observed pre-treatment low grazer biomasses could explain the high MPB concentrations around the oysters reefs unaided by other processes. After experimental treatment, MPB levels around the treatment reef did not recover relative to the control reef until at least April 2015, nor did they form a spatially cohesive patch again until even later[16], while our pigment markers give evidence that grazing levels increased on average post-treatment. Spatial patchiness of a resource can be induced by consumers, especially when their mobility is restricted to refuges between high tides[44]. Predation can both consumptively and non-consumptively disrupt top-down control of primary producers by grazers[45,46]. Crabs such as *Carcinus maenas* and *Hemigrapsus* sp. are important predators of *M. balthica* and *P. ulvae*[47]. For example, Rafaelli et al.[47] enclosed varying natural densities of *C. maenas* (10–15 mm carapace width) on an estuarine intertidal mudflat and showed that crab predation could significantly lower *P. ulvae* density and can result in size structuring of prey populations, as we observed. In Chesapeake Bay, blue crabs, *Callinectes sapidus*, adjust their search patterns to increasing clam densities, i.e., a Type III functional response[48], probing the substrate with the chemosensory and tactile setae on their legs. Crabs thus encourage a deeper burial depth of infaunal clams (>15 cm)[48]. Meanwhile, predation by common shorebird species (e.g., oyster catchers, *Haematopus longirostris*) and benthic fishes (e.g., flounder, *Platichthys flesus*) had no significant effect on grazing prey densities, including *P. ulvae*, in caged exclusion experiments[49]. On intertidal exposed mudflats, crabs need shelter during low tide to avoid predation by birds such as Eurasian curlews and oyster catchers[4]. Bivalve reef habitat provides ideal refuge for crabs[50], and can promote these birds by increasing prey abundances[7]. Predation refuge by oyster reefs may also play a more important role in cooler estuaries, while shading becomes more important at lower latitudes[51]. We show evidence that crabs shelter in the reefs at low tide, emerging to forage up to 65 m outwards on the surrounding mudflats at high tide, producing a halo inversely akin to the impacts of grazing fish around coral reefs[39]. This likely explains the low grazer biomass and altered grazer population structures near to the reefs, which may result in a local break from grazing of MPB biomass.

Bivalve biodeposition was not directly measured in this study, and our analyses did not provide clear support for or against local MPB biomass stimulation by bivalve biodeposition. Biodeposition and subsequent remineralisation of mucus-trapped organic and inorganic particles[36], as well as the excretion of ammonium and phosphates[52] may fertilise MPB-rich patches, which have already been described around oyster farms in otherwise uniform sandy and low-MPB substrate[14,15]. Silt content and sediment organic matter also can decrease with distance from intertidal bivalve reefs, with authors emphasising the importance of bivalve bio-deposition in creating these patterns[4,53]. We found no correlation between distance from the reefs and sediment organic content (depths of 0–50 and 50–100 mm) but observed evidence for organic matter concentrating to the seaward side of the reefs (Supplementary Fig. S8[30]). Biodeposition from molluscs may also displace macrofauna[4], although the infaunal species that showed decreased biomass close to our reefs are typically resistant to organic enrichment (i.e., *M. balthica*, *P. ulvae*[54]), favouring the candidacy of other processes to suppress their biomass. Biodeposition would become less important as oysters metabolic rates decrease in winter, when MPB nutrient requirements also decline, and allochthonous organic matter from the nearby Falleron river (500,000 m³ day⁻¹) relaxes nutrient limitation[18,28]. During tidal inundation, turbulence and water currents should distribute excreted nutrients, potentially linking the biodeposition and abiotic hypotheses around these large, complex structures[30].

MPB patterns were unrelated to MGS and organic content in early autumn at scales of tens of metres. However, evidence for abiotic patterning of MPB was greater in winter, presumably because of stronger seasonal turbulence and currents, while lower temperatures and shorter day lengths lower MPB growth, grazing rates and biodeposition. Muddy, MPB-rich patches have been attributed to disruption of local hydrology by oyster farm structures previously[15,18,28]. Donadi et al.[53] showed hydrodynamic forces to be reduced by 20% by mussel reef presence (via mass loss of exposed plaster). Reefs can facilitate siltation, although effects may focus on the coastal wake of reefs, since sediment grain size reflects the rate of water movement and wave action[29,55]. The settlement of fine grains facilitates binding with organic matter and makes resuspension more difficult. Disruption of local hydrology by reefs is therefore viewed as a secondary process that aids in concentrating oyster biodeposition in summer, nearby the reef[18,28].

Relationships between MPB biomass and meiofaunal patchiness are usually studied at finer scales of observation than those studied here, e.g.,[31,38]. Despite spatial structuring in nematode and harpacticoid copepod abundances, meiofaunal abundances were not significantly related to MPB at the scales in our study.

**Table 2 Pre-treatment (September 2013) spatial correlation matrix between environmental variables and biomasses of dominant consumer species, including the facultative suspension feeders M. balthica and S. plana, the grazer P. ulvae and predators R. obtusa and N. hombergii.**

| Variable | Trend | Transform | MPB NDVI (LN transformed) | | Distance from oyster reefs (√) | | Distance from rock (√) | | Putative prey biomass/abundance (see caption) | |
|---|---|---|---|---|---|---|---|---|---|---|
| | | | R | P (df.) | R | P (df.) | R | P (df.) | R | P (df.) |
| MPB NDVI | | LN | - | - | **-0.797** | 0.001 (20.1) | **-0.508** | 0.028 (16.7) | - | - |
| M. balthica biomass g m⁻² | Y | LN | **-0.535** | 0.007 (22.0) | **0.523** | 0.004 (26.9) | 0.218 | 0.265 (25.9) | - | - |
| S. plana biomass g m⁻² | Y | LN | -0.327 | 0.109 (23.2) | 0.416 | 0.056 (19.7) | -0.075 | 0.749 (18.6) | - | - |
| P. ulvae biomass g m⁻² | X+Y | √ | **-0.521** | 0.001 (40.0) | **0.456** | 0.003 (39.4) | 0.238 | 0.057 (62.6) | - | - |
| R. obtusa biomass g m⁻² | X+Y | Spearman | -0.117 | 0.148 (104.8) | 0.109 | 0.171 (106.9) | -0.092 | 0.911 (180.5) | **0.315** | 0.002 (129.5) |
| Nephtys mm m⁻² | | √ | **-0.376** | 0.003 (60.0) | **0.379** | 0.002 (60.3) | 0.183 | 0.122 (71.1) | - | - |
| Nematoda abundance | | √ | -0.23 | 0.17 (35.5) | 0.13 | 0.457 (33.2) | - | - | - | - |
| Copepoda abundance | | LN | -0.08 | 0.437 (84.8) | 0.05 | 0.687 (76.4) | - | - | **0.52** | <0.001 (76.8) |
| Bathymetry (m) | | | -0.033 | 0.755 (90.5) | - | - | - | - | - | - |
| Macrofauna diversity, 1 - D | Y | | 0.19 | 0.095 (74.6) | -0.22 | 0.073 (67.6) | -0.12 | 0.38 (55.9) | - | - |

A significant trend with X and/or Y axis means the variable was detrended prior to normality tests; non-normal data were LN or √ transformed, else correlations were tested by Spearman's rank. Significance tests used an adjustment of the degrees of freedom (df.) in the presence of spatial autocorrelation via Dutilleul et al.[69], usually resulting in non-integer df. values far lower than non-spatial equivalents (df. = 194, n = 196). Significant coefficients are highlighted in bold.

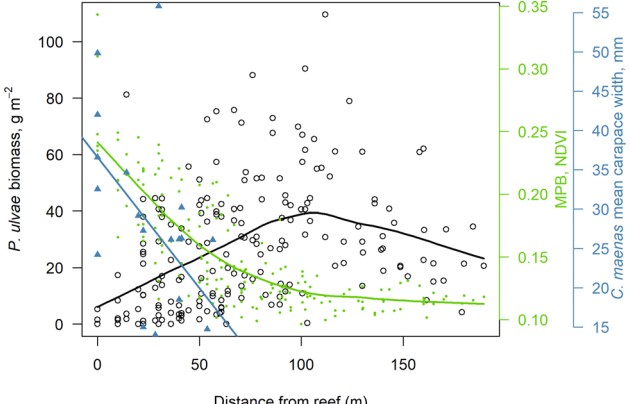

**Fig. 5 The spatial footprint of C. gigas reefs on an index of crab foraging intensity (mean carapace width), the biomass of mudflat grazers, and MPB in the vicinity of 10 s of m.** Crab mean carapace width is shown by blue triangles, right y-axis, with the blue linear regression line $y = 36.6 - 0.33x$ ($P = 0.014$, degrees of freedom = 18, $R^2_{adj} = 0.25$). P. ulvae biomass shown by open black circles, left y-axis, smoothed black curve, which shows a similar pattern to mean individual mass, also for M. balthica (Supplementary Fig. S16). MPB shown by small solid green circles, right y-axis, smoothed green curve. Smoothed curves calculated by first degree LOESS with a smoothing parameter of 2/3.

Meiofauna are nevertheless important players in the community, with carbon turnover levels typically being similar between meio- and macrofauna in North Atlantic mudflats[56].

Ecosystem engineers generally have positive effects on biodiversity, from fine to broad spatial scales[1,38,57]. Whether this also holds for non-native ecosystem engineers is not clear (examples in[9]) although some, including C. gigas, may be functional equivalents to eradicated native species[58,59]. C. gigas can successfully colonise both soft and hard substratum, resulting in increased substrate heterogeneity in both[2,12]. Because of the scarcity of hard substrate in soft sediment systems, engineering effects on these systems have a far higher ecological impact than on hard substrate systems[12]. For every 1 m² sediment surface, Lejart & Hily[12] recorded C. gigas reefs to add 3.9 m² in biogenic substrate. As we document, this facilitates hard substrate species, as well as moderating environmental conditions and providing shelter to these species from fish and seabird predation[60]. In addition, mudflat fauna near to the reefs were slightly more diverse, but were far poorer in abundance and biomass. We show how indirect impacts of the oyster reefs[53] likely operate via a cascade of effects including the facilitation of crab predators[51]. The suppressed infaunal species were important bioturbators[23,61] and had the same biomass in our study as the reefs themselves, hidden under the mud. Observations of positive MPB-macrofauna distribution relationships at a spatial resolution of 100 m[35] may eventually be inverted as oyster reefs encroach larger portions of the mudflat, changing its functional trait distribution[53]. Similar transformations are demonstrated even following the change in habitat complexity from blue mussel to oyster reefs[2,40].

The extent of Crassostrea reefs in temperate regions is predicted to expand in the future[19], which may be welcome as a nature-based means of coastal defence[13], especially if this species effectively restores a habitat previously lost to overexploitation[58,59]. Warming temperatures and gregarious larval settlement cause C. gigas reefs to grow in size, forming still more suitable settlement habitat for larvae[62]. Models of another Atlantic bay predict local decreases in macro-infaunal activity (biomass and respiration) associated with geochemical changes in the sediment[56]. Our study

**Table 3 Summary of study variables and their sampling methodologies.**

| Variable | Sampled on | Methodology | Preservation |
|---|---|---|---|
| MPB NDVI | Raster image | Remotely sensed (Echappé et al.[16]) | NA |
| MPB pigment composition and biomass | L-transect ($n = 25$) | Contact cores sampling (three 2 * 56 mm) | −80 °C |
| Mudflat macrofauna | Random ($n = 196$) | 200 * 200 mm core, sieved (mesh = 1 mm) on site | Field; formalin |
| Meiofauna | Random ($n = 98$) | Two 50 * 25 mm cores (mesh = 40 μm) | Formalin |
| Granulometry (median grain size μm) | Random ($n = 196$) | 100 * 25 mm core; split into 0–50 mm and 50–100 mm depth samples; triplicate subsamples | Freeze-dried at laboratory |
| Sediment organic matter (%) | Random ($n = 196$) | 0–50 mm and 50-100 mm depth samples; Ignition was 1 h at 450 °C. Oven drying was 48 h at 95 °C | −20 °C |
| Distance from nearest reef (m) | Random ($n = 196$) | Euclidean distance calculated. √ transformed. | NA |
| Bathymetry (m) | Random ($n = 196$) | Contour lines were interpolated via inverse distance weighting. Surface was then sampled mathematically | NA |
| Oyster reef biomass | Raster image | Le Bris et al.[17] | NA |
| Reef epifauna | Haphazard ($n = 20$) | Counts from 0.5 * 0.5 quadrats | NA |
| Nets for mobile fauna | Random ($n = 30$) | Fyke nets at left at core sites over two high tides | NA |

Detail for each in Supplementary Results. Macrofauna and MPB NDVI, pigment composition and biomass detailed in main text. Core sizes = depth * diameter.

shows the potential of oyster reefs to increase predation locally, which can suppress mudflat macrofauna. Therefore, although more food is provided via MPB, increased risk of predation may reduce local transfer to benthic deposit feeders and grazers [cf. 56]. Macrofaunal suppression may also have cascading negative impacts on other predators. *P. ulvae* is an important prey for many shorebirds, including redshank, Eurasian curlew, Eurasian oystercatcher, and shelduck, and for commercially important fish species, such as flounder, *P. flesus*[49]. Meanwhile, Eurasian curlews, Eurasian oyster catchers, and also European herring gulls may be able to feed on the crabs facilitated on reefs[7]. Widespread redirection of mudflat trophic flows may thus harm certain nearshore fisheries and affect bird conservation efforts.

Increases in MPB biomass concentrating around oyster reefs may also represent an opportunity for oyster farmers. MPB may comprise up to 50% of water column microalgal biomass because of tidal suspension[63], potentially making major contributions to the diet of cultivated oysters, e.g., up to 50%[64]. However, the importance of this opportunity may depend on water renewal rates of the bay in question[37]. Oyster farmers are increasingly looking to wild stocks of *C. gigas* to replenish aquaculture stocks lost to mass mortality events[17], such as disease outbreaks, that are becoming more frequent. Our results suggest that the spread of wild oyster reefs and increased manipulation of established reefs may strongly influence MPB biomass on which they feed. Understanding the balance of processes linking oyster reefs and MPB biomass (and wider community components), such as those we describe, may allow coastal management, conservationists, farmers, and fishermen to anticipate and mediate local impacts of wild oyster reef spread or removal.

## Methods

**Study area and date**. In the north-west of France, the macrotidal Bourgneuf Bay (1°-2° W, 46°-47° N; total area ~340 km$^2$; Fig. 1) has an intertidal zone largely dominated by mudflats (exposed surface area ~100 km$^2$). Bourgneuf Bay is situated south of the Loire estuary and is open to the sea along 12 km from the west to the north-west. *C. gigas* aquaculture here is of national importance and wild *C. gigas* reefs can account for over double the biomass of their farmed conspecifics[65]. Analysis of satellite observations covering 30 years of MPB biomass in the bay confirmed the co-occurrence of high MPB biomass with wild oyster reefs and cultivated stocks[16] (Supplementary Methods: Wider Situation of the Reefs). Two small (each > 750 m$^2$) wild *C. gigas* reefs and their immediate surroundings (10–100 m) in the north of Bourgneuf Bay were deemed suitable for experimental manipulation (yellow and orange regions in Fig. 1). Méléder et al.[18] described MPB biomass as mostly concentrating around the 2 m isobath, the Falleron river channel (closest point ~400 m NNE from the eastern reef), and oyster farms. Covering this isobath, we superimposed a 350 * 350 m grid (12.25 hectares) to cover the two wild

oyster reefs, orientated so that the 'Y' axis runs parallel to the slope of bathymetry (Fig. 1). The grid was split regularly into 49 'grand-cells' of 50 * 50 m (i.e., $n = 49$) and each of those split into 25 cells of 10 * 10 m (i.e., $n = 1225$; Fig. 1). Four cells per grand-cell were chosen randomly for the sampling of meiofauna, granulometry, OM (see Table 3 for specific methodology), and macrofauna. Of these cells, only every second cell was processed for meiofauna because of time constraints in assessing their abundance.

Although there were only two oyster reef complexes ('reefs', hereon) in this study, multiple sampling cells fell on, or in close proximity to, each reef, so that each reef had many potential (though not independent) distance decay transects running from it capturing natural variation in spatial structure[66]. Comparing the ecological change following the experimental burning of oyster reefs (described below) against ecological change occurring at these two reefs over the previous 25 years[16] also allowed us greater confidence to disentangle the treatment effects from typical variation. Through the centres of five grand-cells to the south of the extent, a transect forming an 'L' shape (Fig. 1) was sampled every 10 m for in situ MPB pigment composition and biomass. We used these data to complete the remote sensing approach for MPB biomass estimation (see below, *Microphytobenthos*). The western reef was slightly larger than the eastern reef and contained a large rock, 'Roche Bonnet', rising 0.5–1 m from the sediment. Outside the grid, another larger (200 * 80 m) wild oyster reef lies WSW at ~260 m distance from the western reef. The grid was sampled for the variables listed in Table 3 during the winter MPB low and early autumn peak seasons (see also ground-truthing in[16]), on the dates 18-19th September 2013 and 17-18th March 2014, before treatment, and on 7-8th October 2014 after treatment.

**Microphytobenthos**. We mapped MPB biomass by satellite remote sensing, following the method described in detail in Echappé et al. (2018). We used the same long-term record of high-resolution satellite images to analyse the spatial distribution of the normalised difference vegetation index (NDVI), a proxy of MPB chlorophyll a concentration at the sediment's surface[18,67], before and after treatment (individual image details in captions of Fig. 2 and Supplementary Figs. S4–S7). After atmospheric correction (FLASH and US40 aerosol model), the satellite-derived NDVI was validated against associated field measurements ($r^2 = 0.85$, root-mean-square deviation, RMSE = 0.04, $n = 57$, $P < 0.05$; also Echappé et al., 2018). The availability of satellite images at both the desired spatial resolution (≤10 * 10 m) and meeting stringent conditions (cloud-free sky, study area emerged from tide, sun elevation >20°) was limited (Supplementary Results: Additional MPB Images). An optimal image was chosen as representative of MPB biomass patterns per season[16]. The study area would ideally be tidally uncovered for ~2 h before the image was taken, whereupon MPB biomass is concentrated at the sediment surface (Fig. 2). The optimal images met this condition (i.e., Fig. 2).

To complete NDVI maps, in situ MPB pigment composition and biomass were retrieved by HPLC analysis from the 25 triplicates of sediment. These had been sampled using contact-cores to freeze the top 2 mm of sediment in situ with liquid nitrogen, with a metal surface 56 mm in diameter. Biomass was expressed by Chl a concentration (mg m$^{-2}$), and dominance of MPB taxa was broadly assessed by ratio of pigment sources to Chl a: Fucoxanthin (Fuco), Diadinoxanthin (DD), Diatoxanthin (DT) and Chl c for diatoms. The ratio of unknown carotenoids (interpreted as by-products due to the low resolution of their absorption spectra) to Chl a was also analysed for ecological purposes (dominant taxa), whereas grazing pressure was investigated using the ratio of pheophorbid a to Chl a (methodological discussion in[28]).

**Sediment variables**. For laser granulometry, we sampled two depths, 0–5 cm and 5–10 cm, in triplicate at each cell. Each of the triplicate samples was put in a vial with water and sonicated. The particle size distribution was determined on a Mastersizer 3000 with a reporting range 50 nm to 3 mm. We also determined sediment percentage OM at two depths by mass loss on ignition in comparison to the oven-dried original (procedure as described for *Macrofauna*, also Table 3).

**Macrofauna**. We sampled macrofauna by a single 200 * 200 mm (depth * diameter) core per cell. Contents were placed into labelled buckets and sieved onshore (1 mm mesh). Soft-bodied polychaetes were picked out with forceps and preserved in buffered formalin during sieving. All material left on the sieve was bagged and preserved in formalin at the laboratory. Individuals were counted and measured by the longest axis (accuracy 0.1 mm, calipers); the deep-burrowing polychaete *Diopatra biscayensis* was counted by the presence of visible tubes above the sediment. Calibration curves from length to mass per species per season were built by identifying size classes by Sturges rule. Multiple individuals per size class (ideally $n = 100$) were measured to estimate mean organic mass per individual of each size class. Shell matter was physically separated from tissue, before both being dried in aluminium foil cups for 48 h at 60 °C and weighed (g) for tissue dry mass using a mass balance. Dry mass was then incinerated for four hours at 450 °C and reweighed (g; decrease in mass of the aluminium cup was also accounted for), the difference giving the organic matter mass (including residue in the shell matter), or ash free dry weight (AFDW). This number was divided by number of individuals. Calibration curves per species used first order polynomial curves for bivalves, unless numbers of size classes and individuals were small (<5 and <20 respectively), in which case the more restricted power curves were used. High variation in the mass ~size relationship, for gastropods because of issues separating shell from tissue via crushing, and for polychaetes because of their stretchable bodies, meant that power curves were again used. Calibration curves allowed the biomass of each species per sample to be estimated.

Total AFDW biomass estimates for the dominant species (≥100 individuals per season) over the study extent (0.1225 km²) were extrapolated from sample means. Biomass estimates for *C. gigas* were calculated in Le Bris et al.[17], by a combination of remote sensing and field observations, before upper and lower range estimates were converted from wet weight to AFDW using the conversion equation for *Crassostrea virginica*[68].

To estimate the distance-decay of predation from oyster reefs, the control reef was revisited over 24-25th July 2017. The original sampled cells were binned into distance classes from the reef i.e., 0–10, 10–20, 20–30 m, etc up to 60 m. Maintaining an even spread by distance class, 30 cells were randomly selected that were visited to deploy standardised fyke nets (nylon multifilament, mesh size ~3 mm, approx. 550 × 220 mm L × W, mouth diameter ~60 mm) over two low tides. Nets were then collected and their contents taken ashore for identification and measurement. Linear regression was used to estimate mean crab carapace width from distance from the reefs, with fishing failures (collapsed or lost nets) omitted. Mean carapace width, integrating both abundance and body size, gives an indicator of crab foraging intensity and depth, since deeper, larger infaunal prey are only available to larger individual crabs[48]. Our distance-decay model assumes that crabs were captured while foraging and must return to the reefs during low tide to avoid becoming prey themselves to birds[4]. High variance and a low sample size precludes the use of curvi-linear methods e.g., a polynomial regression term.

During the 2017 visit, we recorded reef-dwelling fauna by twenty haphazard 0.5 * 0.5 m quadrat samples over the control reef. Samples recorded the percent cover of stationary organisms, including *C. gigas*, counted individuals of mobile organisms, and measured carapace widths of crab species.

**Meiofauna**. We sampled two replicate 50 * 25 mm (depth * diameter) cores per cell for meiofauna, which were preserved in formalin onshore. In the laboratory, samples were sieved on a 40 μm mesh, re-suspended, thrice centrifuged, and extracted in a Ludox solution (density 1.31 g cm⁻³). The fraction was then sorted and counted by microscope down to large taxonomic groups (Nematoda, Cnidaria, Kinorhycha, Polychaeta, Oligocheta). We exclude Foraminifera and Ostracoda because they have external shells (a.k.a. tests) that accumulate in sediment, which make estimates of abundances less reliable. The huge amount of work counting meiofauna precluded the collection of any more than autumn 2013 data only,

**Before-After Control-Impact (BACI) setup**. To sacrifice the non-native oysters over the >750 m² reef expanse, the most eastward of the two oyster reefs was entirely covered with a ~30 cm thick layer of straw and burned over two consecutive low tides (16-17th July 2014; termed the 'treatment'). The wind was allowed to spread the fire rather than using any other additional substance (Supplementary Fig. S18 shows photos of the burning procedure and the effects on the oyster reef). A revisit to the reefs on the 19th July 2014 visually confirmed high-to-complete oyster mortality on the treatment reef only and that the reef shell base remained largely unmodified by the burning (e.g., Supplementary Fig. S18). The larger, westward reef remained as a control. Post-treatment sampling was 81 days afterwards to allow other natural processes that were disturbed by the burning to recover, although lingering artefacts of the burning cannot be ruled out (see discussion; also, Supplementary Results: The timescale of the impact).

**Statistics and reproducibility**. All our analyses and mapping techniques assumed at least second order stationarity (i.e., mean, variance and covariances constant over space). However, a bathymetric trend was expected in biological and sediment variables along the *y*-axis (Fig. 1), which could introduce spurious correlations, so linear trends were tested for by simple linear regressions. The residuals of significant trends were retained as the detrended variable (trends listed in Results), unless the trend was directly incorporated in the spatial prediction model (e.g., simple Kriging). Normality was also assessed for each variable by histograms and Shapiro–Wilk tests prior to parametric tests, leading to natural log or square root transformations where necessary (e.g., NDVI was LN transformed for early autumn 2013 but not for winter 2014; transformations listed in Results). If transformations failed to give an approximately normal distribution, non-parametric tests were used on the raw data (e.g., Wilcoxon's test for pigment ratio). Distance measures were square root transformed.

To test for correlations between variables and MPB, satellite NDVI images were sampled to the 196 sample cells for consistency with other variables. *P*-values for correlation coefficients were calculated using the adjusted degrees of freedom from the Dutilleul et al.[69] method, to counter the elevated risk of Type I statistical error with spatially autocorrelated data. Global (i.e., over the entire study grid) changes in variables from 2013 to 2014 early autumn sampling were tested by paired Wilcoxon tests. All tests were two-sided.

We used structural equation modelling (SEM) to assess which hypothesised ecological process(es) best explained the observed data[70]. SEM has previously been used to explain the impacts of oyster reefs on bird occurrences[4], with a limited assessment of macrofauna. We contrast our main hypotheses based on pathway coefficients of four similarly-structured models, with 2013, 2014, and meiofauna data modelled separately to accommodate differences in the availability of variables and sites. The meiofauna model was dedicated to hypothesis (4), which expected a direct negative link between MPB and grazing, the latter inferred from nematode abundance, which could be mediated by copepod predation (inferred from abundance) and MGS. This was expected regardless of treatment impact, so hypothesis (4) was tested in the pre-treatment 2013 data only, which also halved the large workload involved in counting meiofauna. The other three models all tested hypotheses (1–3). Macrofaunal grazing was modelled as a latent variable approximated by the biomasses of *M. balthica* and *S. plana*. Variables are 'latent' when they are not directly observed but rather inferred from measured variables. The square root of distance from the reefs, reversed so that highest values were closest to the reefs, represented the non-linear distance decay of different processes according to each hypothesis. Hypothesis (1), local enrichment via *oyster biodeposition*, was not directly evidenced but emphasised a direct positive effect of distance from reefs on NDVI and OM. The other hypotheses emphasised indirect effects of distance from reefs on NDVI, via MGS (and OM) for the *abiotic* hypothesis (2), via macrofaunal grazers for the *predation* hypothesis (3), or via meiofaunal grazers for the *meiofaunal* (4) hypothesis. Hypothesis (3) expected grazer biomass to have a negative effect on MPB, while MGS, OM, and bathymetry were also allowed to affect distributions of both grazer groups.

Following the experimental treatment, we expected the link between distance from treatment reef (only) to NDVI to become insignificant under (1) or (3), visualised as a localised decrease in MPB relative to that surrounding the control reef. This was hypothesised because (1) killing the oysters halts *biodeposition*, or (3) because grazer abundance is no longer checked by predators, which were removed in the burning, although a recolonisation of the reef by predators may have occurred by the time of sampling (not recorded). Under (2), the experimental treatment is hypothesised to have no effect on MPB or sediment characteristics at the resolution of this experiment i.e., tens of metres (assuming the reef shell base remains intact; see Supplementary Fig. S18). Further details and SEM equations in Supplementary Methods: SEM. We use R package *lavaan* (v0.6-7)[71]. Parametric assumptions tested for as above. As with our correlation analyses, we account for spatial autocorrelation by correcting SEM effective sample sizes for Moran's I values using function 'lavSpatialCorrect' by Jarrett Byrnes (https://github.com/jebyrnes/spatial_correction_lavaan). This function corrects SEM standard error and significance values for spatial dependence. We assessed model fit by the comparative fit index (CFI), which ideally equals one, and the root mean square error of approximation (RMSEA), which ideally equals zero.

**Mapping of ecological and environmental variables**. The spatial structure in abundance, biomass, diversity, and environmental variables was mapped by geostatistics with the R package *gstat* (v2.0-6)[72]. We used Simpson's 1 - D for diversity because it gives more weight to the common species we are interested in. Correlation strength between pairs of geographical samples (i.e., autocorrelation, characterised by semivariance) typically weakens with increasing distance between sample pairs (lag distance), a relationship plotted as the semivariogram. These are linear methods, so variables were detrended and normalised as explained above, except for the bivalve *Scrobicularia plana* biomass (Supplementary Methods: Geostatistical Details). Variograms used spherical and nugget models, which are well supported in ecology, firstly fitted by eye before secondarily using a weighted least-squares fitting algorithm to optimise sill and range estimates. The semivariograms were then used for simple Kriging interpolation, incorporating any significant trend coefficients back into the Kriging models as independent variables. Informing the Kriging model of these known trends (e.g., *y*-axis was expected to

model bathymetry) allows prediction maps to not only reflect local variation but also natural gradients across the study extent. Analysis and mapping of biomass patterns were concentrated on macrofaunal species that contributed >1% of the total abundance. All mapping and analyses were performed in the statistical computing environment R (v4.0.2)[73].

**Reporting summary**. Further information on research design is available in the Nature Research Reporting Summary linked to this article.

## Data availability
The datasets analysed during the current study are available in the Zenodo repository[74], https://doi.org/10.5281/zenodo.5902388.

## Code availability
All used software was publicly available and described in the Methods. No other custom code was used besides the function 'lavSpatialCorrect' by Jarrett Byrnes (https://github.com/jebyrnes/spatial_correction_lavaan).

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

## Acknowledgements

We would like to thank all the courageous volunteers from our research teams (see Supplementary Wider Acknowledgements) who came in all weather conditions to put their feet and hands in the mud, without whom these field campaigns would not have been possible. The oyster experiment and field campaigns were designed and conducted in the framework of the COSELMAR project funded by the Région Pays de la Loire. We thank CNES's ISIS Program for the use of SPOT satellite products. SPOT6-7 data were acquired by Airbus Defence and Space in the frame of the MyGIC project. We also thank the THEIA and CESBIO teams for the provision of SPOT Take5 (CNES) and Landsat (USGS) data. CJR was supported later by Deutsche Forschungsgemeinschaft grant number AB 109/11-1.

## Author contributions

C.J.R. wrote the manuscript, supported by B.C. and A.B.'A. B.C. and A.B.'A. conceived the project idea. P.D., S.F.D. and L.Bac. managed the organisation of samples. C.J.R., V.M. and C.E. analysed the data. C.J.R., P.D., L.Bac., L.Bar., S.F.D., C.E., P.G., B.J., V.M., P.N., V.T., D.Z., N.Z., A.B.'A. and B.C. contributed to fieldwork and editing the manuscript.

## Funding

## Competing interests

The authors declare no competing interests.
