## [Peer Review File · Communications Biology]

Reviewers' comments:

Reviewer #1 (Remarks to the Author):

This paper seeks to discriminate among multiple alternative hypotheses to explain gradients in micro-phytobenthos (MPB) around oyster reefs. MPB is quantified using remote sensing methods and related to key variables sampled in the field by the authors using structural equations modelling. While the paper includes some very interesting and impressive (in terms of spatial intensity) data, I have some concerns about the experimental design (specifically the lack of replication of control or sacrificed reefs, and the single after time) and the interpretation of results given that the significant artifacts that may have been induced by oyster sacrifice and are not controlled. Furthermore, important details are missing regarding how the burn (to sacrifice the oyster reef) was conducted, and what aspects of the reef it influenced (did it just kill oysters or did it modify the shell base and hence the physical structure of the reefs as well). Additionally, while the paper is likely to be of interest to marine ecologists, at present I do not feel that it is broadly enough framed to appeal to a more general biological audience.

I expand on my key concerns below:

1. As noted by the authors on line 433 - the design is pseudoreplicated – there is only one control and one sacrificed reef. Without replication of at least the control (remaining reef), it is impossible to disentangle effects of oyster reef sacrifice from the effects of other environmental changes that may exhibit a spatial gradient across the mudflat.
2. It is also unfortunate that multiple sampling times were not included following sacrifice of the oyster reef. The 'before' data suggest that there is strong seasonality and/or temporal variation in the system. Hence, including only one 'after' time weakens the inferences that can be made.
3. The process of sacrificing the reef involved covering in straw and burning. The experimental design did not include any procedural control for this method of oyster sacrifice, which in and of itself may induce biological change. Without this, the design is unable to partition effects of oyster sacrifice from experimental artefacts of the method of sacrificing oysters. I also think a few more details are needed regarding what exactly the burial in straw and burning entailed (i.e. how much straw, how was it ignited, how long did the fire burn for etc). This might assist in ascertaining what experimental artefacts may have been introduced.
4. Some clarification is needed on what the net effect on the oyster reef 'sacrifice' was. In addition to killing the oysters, did this have any influence on the shell matrix of the reef? Did dead shell material remain? If shell material remained, were the shells disarticulated and provide a differing habitat matrix to the live reef? Such details are necessary for evaluating the logic of the predictions, and the validity of the interpretation.
5. I am not sure that I follow the authors' logic on lines 541-543. This might be because I am somewhat unclear on whether oyster sacrifice involved only death of oysters, or alteration of the shell base of the reef as well? If the shell base was also modified (which I assume might be the case given burning) surely hydrodynamics would also be modified – and hence sediment characteristics as a result? So I am not convinced the prediction would be for no change in MPB from before to after for hypotheses 2? I am also not sure why the prediction for hypothesis 4 would be no change in MPB? Surely any changes in predator assemblages, or in sediment grain-size would influence meiofauna and hence grazing activity?
6. While it is indeed true that Pacific oyster aquaculture has resulted in the establishment of wild populations of the species outside their native range, I think the ramifications of this project are somewhat broader than this. There is rapidly increasing investment and effort in restoring oyster reefs in Europe, the USA and Australasia, and their use in nature-based coastal defense. It is important to understand how such activities might influence sediment processes and transform the sedimentary ecosystems in which they are placed. Other studies have found a high degree of functional redundancy between native and non-native oysters, such the processes investigated here might also apply to native oyster reefs.

7. On a similar note, if this paper is to appeal to a broader audience, it seems that it might be better to frame it a bit more broadly around ecosystem engineers and direct and indirect effects, perhaps also drawing on analogous examples in terrestrial ecosystems.

8. I would like to see a little more justification of the selection of the specific predictor variables. For example, on what basis were crabs selected as the dominant predators? Are there species of benthic feeding fish that shelter in the oyster reef and might perform a similar role in controlling key grazers? The authors mention that the only other abundant species caught in the nets was a shrimp (that did not display spatial patterns with respect to the reefs) – but I wonder if there are fish species that utilize the reefs, which may not be captured by this sampling method?

9. Estimation of predator foraging based on the number and size of individuals trapped in fyke nets seems a bit simplistic. How do the authors know that crabs were actually foraging at a particular distance from the reefs and not simply moving through the habitat? I'd like to see some more justification for this approach over more direct assays of predation rate.

Reviewer #2 (Remarks to the Author):

The manuscript by Reddin et al. is a very interesting manuscript looking at the processes linking MPB with oyster reefs. Multiple hypotheses are formulated and appropriate experiments have been conducted to test these. The conclusions of the study support a combination of the hypotheses depending upon seasons and conditions and this is well discussed. Some minor comments which are mostly related to the writing are detailed below

Abstract

Many strong yet vague statements which do not add meaning : e.g. L27 , L29
Aim would be useful here

Introduction

The first sentence gives the idea that because they are a dynamic system they would not normally provide ecosystem functioning with the 'yet'.

L46 what does changing nutrients mean ?

L64 how is patchiness assimilated ?

L94 is it correct that sampling was done before and after a before and after experiment ? I am not sure I understand

Methods

L451 details on the satellite images used would be useful

I am not an expert in SEM thus I will not be able to comment on this

Reviewer #3 (Remarks to the Author):

In the manuscript entitled "Extensive spatial impacts of oyster reefs on an intertidal mudflat community via predator facilitation" the authors investigate effects of a Pacific Oyster reef on the surrounding benthic community. They use structural equation models to test four hypotheses describing different ecological mechanisms on how the oyster reef affects trophic interactions and resource supply resulting in higher microphytobenthos (MPB) community biomass surrounding the reefs. The models mainly support the hypothesis increased MPB biomass is related to distance from the reef which the authors use as a proxy for crab predation who usually find shelter within

oyster reef structures. The SEM analyses are complemented by a number of correlative and spatial analyses.

I found the manuscript to be generally well written and clearly structured with comprehensive introduction and discussion sections. I find the research question interesting and timely as with increasing rates of invading species in the mid and higher latitudes their consequences for ecosystem functions are of equally high importance. I am not entirely familiar with the tidal flat area literature on this topic so cannot judge how novel the author's findings are.

I do have a few comments regarding the statistical analysis.

(1) The authors analyse spatially explicit data taking into account spatial autocorrelation in most of their analyses except for the SEM analyses. Why? Spatial autocorrelation can (and should) be incorporated into SEM analyses (e.g. `lavSpatialCorrect` function available on GitHub). Alternatively, a piecewise SEM ('`piecewiseSEM`' package) approach without using latent variables would also be possible and allow for the incorporation of spatial dependencies.

(2) Is Distance used as manifest variable for 'excretion' in models (A) and (B) and also used as manifest variable for 'crab predation' in models (E) and (F)? If that is the case, it wouldn't allow for testing two different hypotheses since a distinction between excretion and predation by crabs is statistically impossible regardless if their effects are analysed in two different models. Further if distance from the reef is associated with two different ecological mechanisms, i.e. reduced grazing pressure on MPB due to crab predation and nutrient input by oyster deposition, it does not seem to be a very reliable manifest variable for either latent variable. In that case I would implement a model without latent variables, i.e. use distance from the reef and relate it directly to the other measured variables. Potential ecological interpretation could then be discussed.

(3) Is there a reason for testing all 4 hypotheses separately? The nice thing about SEM is that one can test several hypotheses in one model simultaneously taking into account the direct and indirect effects of different explanatory variables. The different hypotheses would be represented and tested by links in the model structure and one wouldn't have to repeatedly use the same data to test several hypotheses.

Other than that I only have a few additional comments:

I.83: Maybe rather use "anthropogenic" than "artificial"?

L.208 and I.378-9: Maybe I missed it, but what about post-treatment faunal diversity patterns? Were they different or the same as pre-treatment? Just wondering about whether the macrofaunal dominance patterns are actually caused by crab predation. As far as I know strong dominance is common in macrofaunal communities with different species dominating between years.

I.288-289: Maybe, but a distinction between the effects of crab removal and oyster biodeposition is not possible given the data and analysis.

I.332-5: As mentioned before, although ecologically this certainly makes sense, I am not sure that the presented analyses lend enough support for this hypothesis or at least their effects cannot be distinguished from grazing pressure release due to more intense crab predation close to the reef.

Reviewers' comments:

Reviewer #1 (Remarks to the Author):

This paper seeks to discriminate among multiple alternative hypotheses to explain gradients in micro-phytobenthos (MPB) around oyster reefs. MPB is quantified using remote sensing methods and related to key variables sampled in the field by the authors using structural equations modelling. While the paper includes some very interesting and impressive (in terms of spatial intensity) data, I have some concerns about the experimental design (specifically the lack of replication of control or sacrificed reefs, and the single after time) and the interpretation of results given that the significant artifacts that may have been induced by oyster sacrifice and are not controlled. Furthermore, important details are missing regarding how the burn (to sacrifice the oyster reef) was conducted, and what aspects of the reef it influenced (did it just kill oysters or did modify the shell base and hence the physical structure of the reefs as well). Additionally, while the paper is likely to be of interest to marine ecologists, at present I do not feel that it is broadly enough framed to appeal to a more general biological audience.

#We thank the reviewer for their positive comments and have addressed several comments by aiming to be upfront with the reader about the evidence we present and the issues involved with their interpretation.

I expand on my key concerns below:

1. As noted by the authors on line 433 - the design is pseudoreplicated – there is only one control and one sacrificed reef. Without replication of at least the control (remaining reef), it is impossible to disentangle effects of oyster reef sacrifice from the effects of other environmental changes that may exhibit a spatial gradient across the mudflat.

#We respect the reviewer's intentions here and the need for maximal replication in ecological experiments, which is clear from the 196 sample stations we used in the vicinity of the oyster reefs. However, the large spatial scale of this experimental manipulation (each reef > 750 m², sampling to distances >150 metres away from each reef in multiple directions) and high sampling intensity already stretched available resources and time (i.e. the low tide sampling window and sample processing).

Even multiple field controls can be, and usually are, pseudoreplicated (e.g. in space and time). An advantage of our approach, as we mention in the methods (lines 442-445), was that “each reef had many potential (though not independent) distance decay transects running from it capturing natural variation in spatial structure”. Furthermore, the temporal context of the experiment was set by Echappe et al. 2018, who calibrated the temporal response in MPB at these two reefs to variation over 25 years preceding the experiment. They showed that the average difference in NDVI between the two reefs was significantly greater after the treatment than in the past 25 years. Meanwhile the spatial structure of MPB around the treatment reef became more disaggregated, essentially ceasing to be a cohesive patch, after the treatment, a state that persisted into 2015. However, the post-treatment NDVI mean (0.163) was within the bounds of natural variation for that time of year of 0.16 ± 0.02 (mean \pm SD; also Fig. 8 in Echappe et al., 2018). This allows us greater confidence to disentangle the effects of oyster reef sacrifice from the effects of other environmental changes typical for the locality. We have added these comments on the strengths and weaknesses of our design to parts of the results and discussion (lines 291-294, 445-447).

2. It is also unfortunate that multiple sampling times were not included following sacrifice of the oyster reef. The 'before' data suggest that there is strong seasonality and/or temporal variation in the system. Hence, including only one 'after' time weakens the inferences that can be made.

#Although not available for all variables, we provide additional post-experiment NDVI images in Fig. S7, showing the development of MPB patches. As in the response to comment 1, the temporal context of the experimental MPB response was framed by Echappe et al. 2018, who also calibrated the response at these two reefs to variation over 25 years preceding the experiment and to an additional year (2015) after treatment, including seasonal variation. We apologise that the implications of this earlier study were not clear and now bring these conclusions into greater focus now in the discussion (lines 292-296, 314-316).

3. The process of sacrificing the reef involved covering in straw and burning. The experimental design did not include any procedural control for this method of oyster sacrifice, which in and of itself may induce biological change. Without this, the design is unable to partition effects of oyster sacrifice from experimental artefacts of the method of sacrificing oysters. I also think a few more details are needed regarding what exactly the burial in straw and burning entailed (i.e. how much straw, how was it ignited, how long did the fire burn for etc). This might assist in ascertaining what experimental artefacts may have been introduced.

#We now include greater detail on the burning procedure. We covered the treatment reef entirely with a thick layer of straw and allowed the wind to spread the fire rather than using any other additional substance. We did this on low tides on 16th and 17th July 2014. Although the precise quantity of straw and burning duration were not recorded, we now include additional photos in the SM (Fig. S18) to allow its estimation and have added details to the methods (lines 544-548).

4. Some clarification is need on what the net effect on the oyster reef 'sacrifice' was. In addition to killing the oysters, did this have any influence on the shell matrix of the reef? Did dead shell material remain? If shell material remained, were the shells disarticulated and provide a differing habitat matrix to the live reef? Such details are necessary for evaluating the logic of the predictions, and the validity of the interpretation.

#We now include additional photos in the SM (Fig. S18) that show the process and its immediate and lasting effects on the oyster reef. After the burning, the oysters shells are seen to be physically fully intact. Dead shells now permanently gape instead of closing during low tide, as in life.

5. I am not sure that I follow the authors' logic on lines 541-543. This might be because I am somewhat unclear on whether oyster sacrifice involved only death of oysters, or alteration of the shell base of the reef as well? If the shell base was also modified (which I assume might be the case given burning) surely hydrodynamics would also be modified – and hence sediment characteristics as a result? So I am not convinced the prediction would be for no change in MPB from before to after for hypotheses 2? I am also not sure why the prediction for hypothesis 4 would be no change in MPB? Surely any changes in predator assemblages, or in sediment grain-size would influence meiofauna and hence grazing activity?

#The reef shell base remained intact after the burning so that hydrodynamics at the scale of 10s m was not expected to change from 2013 to 2014. We have added this assumption and its photographic support in the methods. Hypothesis 4 is primarily tested by the SEM model using 2013 data only, because of the huge work in counting meiofaunal samples (which is now mentioned in the methods, L541-542), so we have omitted its mention here in the BACI section of the methods.

6. While it is indeed true that Pacific oyster aquaculture has resulted in the establishment of wild

populations of the species outside their native range, I think the ramifications of this project are somewhat broader than this. There is rapidly increasing investment and effort in restoring oyster reefs in Europe, the USA and Australasia, and their use in nature-based coastal defense. It is important to understand how such activities might influence sediment processes and transform the sedimentary ecosystems in which they are placed. Other studies have found a high degree of functional redundancy between native and non-native oysters, such the processes investigated here might also apply to native oyster reefs.

#We thank the reviewer for these suggestions. We now add statements to the introduction, “that are of interest for nature-based coastal defense”, and discussion, “which may be welcome as a nature-based means of coastal defence, and if this species effectively restores a habitat previously lost to overexploitation”.

7. On a similar note, if this paper is to appeal to a broader audience, it seems that it might be better to frame it a bit more broadly around ecosystem engineers and direct and indirect effects, perhaps also drawing on analogous examples in terrestrial ecosystems.

#Following this recommendation, we have restructured the opening paragraph to begin discussing ecosystem engineers.

8. I would like to see a little more justification of the selection of the specific predictor variables. For example, on what basis were crabs selected as the dominant predators? Are there species of benthic feeding fish that shelter in the oyster reef and might perform a similar role in controlling key grazers? The authors mention that the only other abundant species caught in the nets was a shrimp (that did not display spatial patterns with respect to the reefs) – but I wonder if there are fish species that utilize the reefs, which may not be captured by this sampling method?

#Our fyke nets also caught three individuals of the flatfish *Solea solea* but their intertidal foraging is unlikely confined to the reef surroundings (Le Pape et al. 2013). This, and the sentence, “Fyke net and reef quadrat sampling over high tide identified the crab *C. maenas* as the most abundant and likely predator of infaunal bivalves”, have been added to the results (L248-252). Until then, we have checked that we do not mentioned any assumption of which predator is responsible for the infaunal patterns. The references documenting the strong potential influence of *C. maenas* on northwest European soft bottom infauna then support the likelihood of their role in determining the observed infaunal patterns.

9. Estimation of predator foraging based on the number and size of individuals trapped in fyke nets seems a bit simplistic. How do the authors know that crabs were actually foraging at a particular distance from the reefs and not simply moving through the habitat? I’d like to see some more justification for this approach over more direct assays of predation rate.

#In the methods we have now made explicit that our approach “assumes that crabs were captured while foraging” (line 527). While this assumption is certainly debatable, crabs must forage to acquire food, and their feeding behaviours for infauna on soft bottoms are well referenced in our discussion. We now also mention the term ‘foraging intensity’ less, preferring to state the original variable (e.g. mean carapace width) in the results and leave interpretation to the discussion.

Reviewer #2 (Remarks to the Author):

The manuscript by Reddin et al. Is a very interesting manuscript looking at the processes linking MPB

with oyster reefs . Multiple hypothesis are formulated and appropriate experiments have been conducted to test these. The conclusions of the study support a combination of the hypotheses depending upon seasons and conditions and this is well discussed. Some minor comments which are mostly related to the writing are detailed below

#We thank the reviewer for their constructive comments.

Abstract

Many strong yet vague statements which does not add meaning : e.g. L27 , L29
Aim would be useful here

#Potentially vague terms have been replaced and an aim has been added to the abstract.

Introduction

The first sentence gives the idea that because they are a dynamic system they would not normally provide ecosystem functioning with the 'yet'.

#Changed to 'and'. Apologies, the intention was 'dynamic yet reliable'.

L46 what does changing nutrients mean ?

#'dynamics' moved forward for clarity.

L64 how is patchiness assimilated ?

#Reworded from 'its patchiness'.

L94 is it correct that sampling was done before and after a before and after experiment ? I am not sure I understand

#The reviewer is correct. Amended as suggested.

Methods

L451 details on the satellite images used would be useful

#We thank the reviewer for pointing this out and have now added details of the individual images to each caption of figures S4-7.

I am not an expert in SEM thus I will not be able to comment on this

Reviewer #3 (Remarks to the Author):

In the manuscript entitled "Extensive spatial impacts of oyster reefs on an intertidal mudflat community via predator facilitation" the authors investigate effects of a Pacific Oyster reef on the surrounding benthic community. They use structural equation models to test four hypotheses describing different ecological mechanisms on how the oyster reef affects trophic interactions and resource supply resulting in higher microphytobenthos (MPB) community biomass surrounding the reefs. The models mainly support the hypothesis increased MPB biomass is related to distance from the reef which the authors use as a proxy for crab predation who usually find shelter within oyster reef structures. The SEM analyses are complemented by a number of correlative and spatial analyses.

I found the manuscript to be generally well written and clearly structured with comprehensive introduction and discussion sections. I find the research question interesting and timely as with increasing rates of invading species in the mid and higher latitudes their consequences for ecosystem functions are of equally high importance. I am not entirely familiar with the tidal flat area literature on this topic so cannot judge how novel the author's findings are.

#We thank the reviewer for their supportive comments and for the very useful criticism of the SEM, which has led to an improved manuscript.

I do have a few comments regarding the statistical analysis.

(1) The authors analyse spatially explicit data taking into account spatial autocorrelation in most of their analyses except for the SEM analyses. Why? Spatial autocorrelation can (and should) be incorporated into SEM analyses (e.g. `lavSpatialCorrect` function available on GitHub). Alternatively, a piecewise SEM ('piecewiseSEM' package) approach without using latent variables would also be possible and allow for the incorporation of spatial dependencies.

#We thank the reviewer for bringing this code to our attention and have now implemented it into the SEM and updated the results (which remain largely the same) and figure 3. This addition is now described in the methods (lines 587-591).

(2) Is Distance used as manifest variable for 'excretion' in models (A) and (B) and also used as manifest variable for 'crab predation' in models (E) and (F)? If that is the case, it wouldn't allow for testing two different hypotheses since a distinction between excretion and predation by crabs is statistically impossible regardless if their effects are analysed in two different models. Further if distance from the reef is associated with two different ecological mechanisms, i.e. reduced grazing pressure on MPB due to crab predation and nutrient input by oyster deposition, it does not seem to be a very reliable manifest variable for either latent variable. In that case I would implement a model without latent variables, i.e. use distance from the reef and relate it directly to the other measured variables. Potential ecological interpretation could then be discussed.

#The reviewer is correct here so we have followed their recommendations to remove the latent variables associated with distance from reefs and to subsequently discuss potential ecological interpretation.

(3) Is there a reason for testing all 4 hypotheses separately? The nice thing about SEM is that one can test several hypotheses in one model simultaneously taking into account the direct and indirect effects of different explanatory variables. The different hypotheses would be represented and tested by links in the model structure and one wouldn't have to repeatedly use the same data to test several hypotheses.

#We agree with the reviewer that some of the models could be combined where multiple hypotheses are compared using the same data. We have therefore restructured our application of SEM as four models (fig. 3). These four cannot be combined further because three use different datasets and one aims to contrast the effect of using a highly collinear variable (distance from both reefs rather than distance from control reef only). The newly freed display space also means that, now, no SEM models are only displayed in the SM: two have now been brought into the main text.

Other than that I only have a few additional comments:

I.83: Maybe rather use “anthropogenic” than “artificial”?

#Amended as suggested.

L.208 and I.378-9: Maybe I missed it, but what about post-treatment faunal diversity patterns? Were they different or the same as pre-treatment? Just wondering about whether the macrofaunal dominance patterns are actually caused by crab predation. As far as I know strong dominance is common in macrofaunal communities with different species dominating between years.

#Unfortunately for 2014, sampling issues for other species (besides *M. balthica*, and *S. plana*) precluded a comparable diversity plot, as stated in the caption of Fig. 4. The effects of crab predation are therefore inferred and we have adapted the text to make this clear. The first line referred to by the reviewer in the results (now L203) has now been changed to reef proximity, rather than crab predation. The second statement in the discussion (L384-386), has the word ‘likely’, so has not been changed: “oysters can indirectly decrease mudflat faunal diversity and their species’ biomasses nearby the reef, likely via a cascade of effects including the facilitation of crab predators”.

I.288-289: Maybe, but a distinction between the effects of crab removal and oyster biodeposition is not possible given the data and analysis.

#We have adjusted the wording in this first discussion paragraph to acknowledge this limitation of the data: “However, it was not clear if this was due to a persistent absence of crabs” (L294).

I.332-5: As mentioned before, although ecologically this certainly makes sense, I am not sure that the presented analyses lend enough support for this hypothesis or at least their effects cannot be distinguished from grazing pressure release due to more intense crab predation close to the reef.

#Following the restructuring of the SEM and reviewer comments, which are correct, the oyster biodeposition section in the results has been changed. The discussion paragraph for this potential mechanism has also been adapted to use more cautious language.

Reviewers' comments:

Reviewer #1 (Remarks to the Author):

The revisions are largely responsive to my previous comments, but I do still have a few remaining concerns:

- Thank you for including photos of the burning, and post-burn reefs. In addition to killing the oysters, it seems that the method of burning left charred straw behind (panel C, Fig. S18) some of which may have been washed away and some of which might have been buried in sediments, the latter potentially adding structure and organic matter. In my opinion the authors still haven't gone far enough to dispel concerns about experimental artefacts of the burning method, which itself may have contributed to ecological change. At the very least it would be good to include a statement in the Discussion that results were not consistent with what might be expected in the case of experimental artefacts.
- The authors now indicate (line 553) that the burn caused high-to-complete oyster mortality. Do they have any data on live vs dead oyster ratio/density from oyster sampling before and after the burn – it would be useful for interpreting the data?
- Similarly, do they have any data on structural complexity of the reefs before vs after the burn. As dead oysters gape, it may be expected that post-burn, the reef has greater complexity, potentially influencing hydrodynamic processes as well as predator refuge. Hence I do not necessarily agree with the assumption that the shell base remained unchanged
- I'm not sure that I agree with the authors as to what constitutes support for the various hypotheses post burn.
 - o For example, I'm not sure why they would necessarily predict some moderate decline in MPB under hypothesis (3). Pre burn, live oysters would provide food and habitat to predators (i.e. *Carcinus* can shelter amongst oysters, and prey and juveniles). Post-burn, the gaping dead oyster shells would presumably provide more habitat to crabs, but less food. So potentially crabs may actually increase the intensity of foraging around the dead reef? So potentially there could be an increase in MPB if there is greater top down control on predators?
 - o Likewise I am not sure I agree that the burn would have no influence on hydrodynamics – given that the structure of the oyster reef base may change (see comment above)

Reviewer #2 (Remarks to the Author):

I find the revised manuscript is improved and the authors did respond to my previous queries adjusting the unclear parts. I believe the manuscript is of interest and nearly ready to be published after some minor checks are done (e.g. axis text on figure 4, small size of axis labels and text on fig 5; use of word 'sacrifice' in L 482 of the tracked version)

Reviewer #3 (Remarks to the Author):

In my opinion Reddin et al have revised their manuscript thoroughly and clarified or corrected most reviewer comments.

I only have a couple more comments:

I still find the description and presentation of the SEM analyses and results confusing. A clear description of which biological hypothesis or known mechanism is associated to which link of the initial SEM structure to be tested and also which of the models presented in Figure 3 deals with which hypothesis would be extremely helpful. Also report that and why you use a latent variable and which indicator variables are associated with it. The authors kind of explain some of it in the methods and some of it in the Figure caption, but I found myself having to go back and forth

between these sections to understand what exactly is tested and shown. For example:

I.595-7: The authors say they expect 'some relationships to change'. Please be more precise. Are the links expected to be the same just less significant or what exactly is the initial model structure for the different models?

I.598-9: 'high MPB in the immediate surroundings of the reef' is represented by the square root of distance? Please be more precise. State clearly whether you expect a non-linear relationship and why and how you incorporate it in the model structure by including which explanatory variables in which equation of the model. If a description seems to take up too much space, report at least the equations for the underlying structural equations.

I.607: I would delete 'relatively' or reformulate.

#Line numbers refer to the version without tracked changes.

Reviewer #1 (Remarks to the Author):

The revisions are largely responsive to my previous comments, but I do still have a few remaining concerns:

- Thank you for including photos of the burning, and post-burn reefs. In addition to killing the oysters, it seems that the method of burning left charred straw behind (panel C, Fig. S18) some of which may have been washed away and some of which might have been buried in sediments, the latter potentially adding structure and organic matter. In my opinion the authors still haven't gone far enough to dispel concerns about experimental artefacts of the burning method, which itself may have contributed to ecological change. At the very least it would be good to include a statement in the Discussion that results were not consistent with what might be expected in the case of experimental artefacts.

#We agree with the reviewer that some residue of the burning is likely to remain 81 days after the burning, when sampling took place, but do not believe this has biased our conclusions. A statement has been added into the methods when describing the burning that "*although lingering artefacts of the burning cannot be ruled out*" (L549). Additionally, the first paragraph of the discussion now includes the statement, "*Although some artefacts of this large experimental manipulation may have remained in the ecosystem sampled in 2014, our multifaceted results were mostly inconsistent with large impacts besides the intended killing of oysters (see Appendix 2)*" (L289-292). The many different variables sampled in this study allow us to investigate many of these possibilities e.g. shallow sediment organic matter rose from 2013 into 2014 but the only spatial structure detected was a negative x-trend (from mean 13% to 10.1%), with highest % organic matter furthest from the treatment reef and no noticeable concentration around it. This latter statement has also been added to the SM, section 'Further procedural details of the reef burning'.

- The authors now indicate (line 553) that the burn caused high-to-complete oyster mortality. Do they have any data on live vs dead oyster ratio/density from oyster sampling before and after the burn – it would be useful for interpreting the data?

#Unfortunately, we do not have any quantitative data on live vs dead oyster ratios before and after the burn, only of the infauna. We have added to the methods that this confirmation of mortality was visual only in nature.

- Similarly, do they have any data on structural complexity of the reefs before vs after the burn. As dead oysters gape, it may be expected that post-burn, the reef has greater complexity, potentially influencing hydrodynamic processes as well as predator refuge. Hence I do not necessarily agree with the assumption that the shell base remained unchanged

#Unfortunately, we do not have any data on structural complexity of the reefs before vs after the burn. We agree that the gaping shells of dead oysters will change the fine-scale water movement but the difference between this and when the oysters are gaping to feed (alongside the pumping of water) on the flood tide is unclear, especially at the spatial resolution here of 10s m. We now specify that the abiotic hypothesis is tested at 10s m resolution, a spatial scale at which the difference between gaping and closed or feeding shells is less likely to be important. A statement to this effect has now also been added to the SM, section 'Further procedural details of the reef burning'.

- I'm not sure that I agree with the authors as to what constitutes support for the various hypotheses post burn.

- o For example, I'm not sure why they would necessarily predict some moderate decline in MPB under hypothesis (3). Pre burn, live oysters would provide food and habitat to predators (i.e. *Carcinus* can shelter amongst oysters, and prey and juveniles). Post-burn, the gaping dead oyster shells would presumably provide more habitat to crabs, but less food. So potentially crabs may actually increase the intensity of foraging around the dead reef? So potentially there could be an increase in MPB if there is greater top down control on predators?

#The reviewer likely agrees that any crabs in the reefs at the time of burning would have been killed. We agree with the reviewer that the impact on grazers observed 81 days after the burning depends on the speed of recolonization of the reef by (large) crabs, which unfortunately we have no data on (now noted in methods). However, we expect large crabs to be more territorial and less mobile, the first arrivals instead to be from smaller individuals that have less influence on the infaunal macro-grazers we sample. This is because small size crabs are less able to reach the burial depth of larger *S. plana* and *M. balthica* and to penetrate hard shells of larger grazers. Large crabs will inevitably recolonise eventually. In 2014, the control reef impact on MPB was largely similar to before the burning (suggesting that MPB facilitation mechanisms were intact). However, that the treatment reef reversed sign to positive suggested that the original MPB facilitation mechanisms were no longer intact in the treatment reef in 2014.

- o Likewise I am not sure I agree that the burn would have no influence on hydrodynamics – given that the structure of the oyster reef base may change (see comment above)

#Please see response to previous comment on structural complexity.

Reviewer #2 (Remarks to the Author):

I find the revised manuscript is improved and the authors did respond to my previous queries adjusting the unclear parts. I believe the manuscript is of interest and nearly ready to be published after some minor checks are done (e.g. axis text on figure 4, small size of axis labels and text on fig 5; use of word 'sacrifice' in L 482 of the tracked version)

#L482 has been changed to “*experimental burning of oyster reefs (described below)*”

The two figures have now been replotted with larger labels and text. The replotting of figure 4A exposed an error in quantifying macrofaunal diversity, which has now been corrected in the current plot Fig. 4A, the correlations in Table 2, main text and supporting figures. Diversity was not a core component of our paper so this had little effect on our conclusions.

Reviewer #3 (Remarks to the Author):

In my opinion Reddin et al have revised their manuscript thoroughly and clarified or corrected most reviewer comments.

I only have a couple more comments:

I still find the description and presentation of the SEM analyses and results confusing. A clear

description of which biological hypothesis or known mechanism is associated to which link of the initial SEM structure to be tested...

#The analytical description of structural equation modelling has been overhauled, including explicit links between links and hypothesised mechanism (see section 'Statistical Analysis').

... and also which of the models presented in Figure 3 deals with which hypothesis would be extremely helpful. Also report that and why you use a latent variable and which indicator variables are associated with it.

#Sentences on latent variables and which model addressed which hypothesis now adapted as: *"We contrast our main hypotheses based on pathway coefficients of four similarly-structured models, with 2013, 2014, and meiofauna data modelled separately to accommodate differences in the availability of variables and sites. The meiofauna model was dedicated to hypothesis (4), which expected a direct negative link between MPB and grazing..."* (L573)

*"The other three models all tested hypotheses (1-3). Macrofaunal grazing was modelled as a latent variable approximated by the biomasses of *M. balthica* and *S. plana*. Variables are 'latent' when they are not directly observed but rather inferred from measured variables."* (L580)

The authors kind of explain some of it in the methods and some of it in the Figure caption, but I found myself having to go back and forth between these sections to understand what exactly is tested and shown.

#The detail of the hypotheses, their relation to the SEM, and expectations following the burning has now been concentrated into a single section, the 'statistical analysis'.

For example:

I.595-7: The authors say they expect 'some relationships to change'. Please be more precise. Are the links expected to be the same just less significant or what exactly is the initial model structure for the different models?

#The description of changes expected following the experimental treatment is adapted to: *"Following the experimental treatment, we expected the link between distance from treatment reef (only) to NDVI to become insignificant under (1) or (3), visualised as a localised decrease in MPB relative to that surrounding the control reef. This was hypothesised because (1) killing the oysters halts biodeposition, or (3) because grazer abundance is no longer checked by predators, which were removed in the burning, although a recolonization of the reef by predators may have occurred by the time of sampling (not recorded). Under (2), the experimental treatment is hypothesised to have no effect on MPB or sediment characteristics at the resolution of this experiment i.e. tens of metres (assuming the reef shell base remains intact; see Fig. S18)." (L592)*

Now in the supplementary: *"The hypothesis (3) treatment response also depended on whether the infauna were affected by the burning and, if they were not, whether predators had successfully recolonised the reef by the sampling date."*

I.598-9: 'high MPB in the immediate surroundings of the reef' is represented by the square root of distance? Please be more precise. State clearly whether you expect a non-linear relationship and why and how you incorporate it in the model structure by including which explanatory variables in which equation of the model. If a description seems to take up too much space, report at least the equations for the underlying structural equations.

#We have now adapted the wording to be more specific about the non-linear distance decay expected for oyster enrichment of MPB, hydrological effects of the reef structure on sedimentation, and crab predation on macrofauna. *“The square root of distance from the reefs, reversed so that highest values were closest to the reefs, represented the non-linear distance decay of different processes according to each hypothesis. Hypothesis (1), local enrichment via oyster biodeposition, was not directly evidenced but emphasised a direct positive effect of distance from reefs on NDVI and OM. The other hypotheses emphasised indirect effects of distance from reefs on NDVI, via MGS (and OM) for the abiotic hypothesis (2), via macrofaunal grazers for the predation hypothesis (3), or via meiofaunal grazers for the meiofaunal (4) hypothesis. Hypothesis (3) expected grazer biomass to have a negative effect on MPB, while MGS, OM, and bathymetry were also allowed to affect distributions of both grazer groups”*. (L193)

#We also now include the notation for the structural equations in the SM Appendix 2.

I.607: I would delete ‘relatively’ or reformulate.

#This part is now reformulated.